# Building Damage Assessment Based on Siamese Hierarchical Transformer Framework

**Yifan Da** †, **Zhiyuan Ji** † and **Yongsheng Zhou** *

College of Information Science and Technology, Beijing University of Chemical Technology, Beijing 100029, China; 2019040191@mail.buct.edu.cn (Y.D.); 2019030331@mail.buct.edu.cn (Z.J.)

* Correspondence: zhyosh@mail.buct.edu.cn
† These authors contributed equally to this work.

**Abstract:** The rapid and accurate damage assessment of buildings plays a critical role in disaster response. Based on pairs of pre- and post-disaster remote sensing images, effective building damage level assessment can be conducted. However, most existing methods are based on Convolutional Neural Network, which has limited ability to learn the global context. An attention mechanism helps ameliorate this problem. Hierarchical Transformer has powerful potential in the remote sensing field with strong global modeling capability. In this paper, we propose a novel two-stage damage assessment framework called SDAFormer, which embeds a symmetric hierarchical Transformer into a siamese U-Net-like network. In the first stage, the pre-disaster image is fed into a segmentation network for building localization. In the second stage, a two-branch damage classification network is established based on weights shared from the first stage. Then, pre- and post-disaster images are delivered to the network separately for damage assessment. Moreover, a spatial fusion module is designed to improve feature representation capability by building pixel-level correlation, which establishes spatial information in Swin Transformer blocks. The proposed framework achieves significant improvement on the large-scale building damage assessment dataset—xBD.

**Keywords:** remote sensing image; building damage assessment; deep learning; two-stage framework; spatial fusion

**MSC:** 53C20

## 1. Introduction

When a serious natural disaster strikes, residential buildings are damaged in high probability, which poses a great threat to property and life [1,2]. According to the statistics, building collapse is one of the main causes of human casualties after natural disasters [3]. Rapid and accurate building damage assessment prior to rescue actions can support effective emergency rescue planning and save more lives [4], and it is essential for Humanitarian Assistance and Disaster Response (HADR) [5]. It has become an indispensable reference for rescue actions after natural disaster strikes nowadays.

Remote sensing has the advantage of acquiring ground target information over a large area and has been widely used to observe disaster areas. In recent years, with the development of remote sensing technology and satellite constellations, remote sensing data have become more easily accessible when disasters occur. High-resolution optical images, synthetic aperture radar (SAR) and LiDAR data are frequently used in the interpretation of disaster-affected areas [6–11]. When a disaster occurs, it becomes an important way to assess the post-disaster situation [12]. High-resolution optical images are more widely used [13] since the real building conditions can be easily interpreted from them, which provides a powerful source of information to assess the extent and scope of the damage.

A number of building damage assessment methods based on high-resolution optical remote sensing images have been proposed.

- According to whether pre-disaster or post-disaster remote sensing images are used, building damage assessment methods can be divided into the following two categories: (1) methods that use only post-disaster images; and (2) methods that use both pre-disaster and post-disaster images. For post-disaster image-based methods, damage assessment is usually conducted by image segmentation [14]. However, the outlines of buildings in post-disaster images may be blurred by the strike of disaster, and the characteristic of the buildings often changes dramatically. Collapsed buildings lose their regular geometric shapes, where the regular texture feature distribution does not apply, resulting in building assessment errors. For the above problems, if pre-disaster image of intact buildings is available and used, it can help locate the damaged parts of buildings [15]. In [16], a two-stage framework is proposed for damage detection from both pre-disaster and post-disaster images. The networks of two stages share the same weight and are responsible for building localization and damage classification, respectively. In this paper, we design our network mainly based on this idea.
- According to the image processing methods used, building damage assessment methods can be divided into the following three categories: (1) manual visual interpretation, (2) traditional machine learning-based methods, and (3) deep learning-based methods. The manual visual interpretation has good specificity and relatively high accuracy, but it takes a lot of time, and the effectiveness of the interpretation depends on human experience and the time spent; thus, it may lead to missing the best time for rescue. Therefore, it is necessary to automatically perform damage assessment from remote sensing images. Many machine learning algorithms with shallow structures have been developed, such as the methods based on Support Vector Machine (SVM) [17] and Random Forest (RF) [18]. These methods have reached relatively high precision within a small number of parameters. Annabella et al. apply SVM classifier on the changed features of buildings, including color, texture, correlation descriptors, and statistical similarity, to detect the destroyed building objects after the earthquake [19]. However, the manually extracted features from remote sensing images by these methods are not sufficiently general, and in most cases, they are only valid for specific situations. These models are difficult to transfer to other geographic areas [20]. Moreover, a high level of a priori knowledge, as well as a large amount of time, is required for the feature extraction. Therefore, it is not practical to apply traditional machine learning algorithms to building damage assessment quickly after the disasters strike.

Deep learning has been extensively developed in recent years, and the methods based on Convolutional Neural Network (CNN) have achieved high-level accuracy on various tasks [21–24] due to its powerful automatic feature extraction capability. Many researchers have made great progress in building damage assessment when introducing CNN. Koch et al. [25] propose a metric-based method with siamese CNN networks for one-shot classification tasks. An energy function of a weighted distance between the twin feature vectors is utilized. The tied parameters between the siamese network allow the same metric computed when two distinct images are fed into the network. Liu et al. [26] use an end-to-end framework to detect damaged buildings from remote sensing images by combining CNN and Recurrent Neural Network (RNN). Nex et al. [27] use multi-level classification instead of binary classification to asses the extent of damage. In [28], for an image patch, all buildings at the edges are occluded to allow the model to focus on the central buildings. Zhan et al. [29] apply U-Net [30], which is originally used for medical image segmentation, on two-phase SAR images for building structure change detection. Based on CNN, U-Net consists of a symmetric Encoder–Decoder in which down-sampling and up-sampling are introduced. U-Net proposes skip connections that enable the integration of low-level features and high-level features, thereby improving the representation of spatial information. With the capability to restore high-resolution information and high-level feature extraction, different kinds of U-Nets have been widely applied to change detection and damage assessment tasks. Yang et al. [31] construct the Recurrent-CNN (RCNN) U-Net, which can extract spatial context and exploit rich low-level

features. RCNN performs region of interest (ROI) segmentation before feature extraction on the ROI instead of the whole image. To exploit the correlation between pre- and post-disaster images, Xiao et al. [32] propose a siamese U-Net to process the task of building segmentation and damage classification simultaneously. However, the actual receptive field of the CNN-based network is much smaller than the theoretical receptive field [33]. In other words, the representation ability of the network is limited. In the process of damage assessment, the limited receptive field constrains the contextual information that the network can utilize, which has a significant impact on the assessment performance.

The Transformer architecture [34] has the advantage of a global receptive field and has the potential to overcome the above problems of CNNs. It initially achieved great success in Natural Language Processing (NLP). The Vision Transformer (ViT) introduces Multi-head Self-Attention (MSA) into vision tasks [35]. Recently, ViT and its variants have shown powerful global relational modeling capabilities and outperform the state-of-the-art CNN models in many tasks [36–38]. Compared to CNN with a limited receptive field, Transformer keeps the sizes of input and output unchanged and effectively captures global contextual information. Chen et al. [39] propose a two-branch ViT with skip connections to learn multi-scale features. It proves that learning features from different scales is also effective in vision tasks. Wang et al. [40] introduce the Pyramid Vision Transformer, which is a unified Transformer backbone for vision tasks with pixel-level prediction that does not require convolution operations. However, there are two challenges in applying Transformer in remote sensing domain, i.e., the various scales of ground objects and the extreme high-resolution images. To limit the high computational complexity of ViT on high-resolution images, Swin Transformer [41] proposes a shifted window mechanism to construct a hierarchical structure and shows great potential in semantic segmentation. The shifted window mechanism significantly reduces the computation cost and make it possible to process high-resolution images. In the field of medical image segmentation, the Swin Transformer architecture shows great performance [42], but its potential has not been confirmed in the field of damage assessment.

Although the shifted window mechanism of Swin Transformer makes the computational complexity linear with the input size, this strategy weakens the global modeling capability of Transformer to some extent, which requires additional spatial information to compensate for it. Moreover, in the building damage assessment based on high-resolution optical remote sensing images, certain classes of damage have highly identical appearances, e.g., no damage and minor damage. Therefore, existing methods use an attention mechanism to address these issues. Fu et al. [43] build long-range correlations through parallel channel attention and position attention. CBAM [44] constructs spatial-level and channel-level attention for adaptive feature refinement. Chen et al. [45] propose a feature extractor with a pyramid spatial–temporal attention module for change detection. In remote sensing images, pixel-level spatial correlation should receive more attention to avoid semantic ambiguity due to the occlusion of ground objects [46]. Therefore, we introduce vertical and horizontal self-attention mechanisms to construct pixel-level spatial correlations.

In this paper, to address the limitation of CNN in global relational modeling, we propose a hierarchical Transformer-based two-stage framework, named SDAFormer, for building damage assessment for the first time. As mentioned earlier, the assessment process is split into two stages to make full use of the semantic information in the pre- and post-disaster images. In Stage 1, a Transformer-based U-Net-like pixel-level segmentation network is used for building localization. Inspired by Residual network [47] and U-Net [30], a symmetric encoder–decoder structure with skip connections is constructed as the segmentation network based on Transformer block. Then, the segmentation results are used to guide the building locations for Stage 2. In Stage 2, damage classification is conducted by using a siamese network. The weights trained in Stage 1 are utilized to initialize the network weights of Stage 2 to improve the efficiency of the training process. In addition, a spatial fusion (SF) module is proposed to enable the network to aggregate global features

in spatial dimensions. The framework is evaluated on the xBD dataset [48] and individual disaster datasets.

The main contributions of this paper can be summarized as follows.

- We propose a Transformer-based two-stage framework for pre- and post-disaster remote sensing image analysis. Based on the siamese U-Net architecture, a pure Transformer-based encoder–decoder structure is constructed for the building damage assessment task instead of CNN.
- To enhance the spatial correlation of global features, a spatial fusion (SF) module is presented. We introduce self-attention in the horizontal and vertical directions to enhance the pixel-level feature representation capability.

The rest of this paper is organized as follows. Section 2 presents the detailed methods. Section 3 presents the data used and experimental results. Section 4 contains the discussion and Section 5 draws conclusions.

## 2. Methods

### 2.1. Overall Architecture

The overall architecture of the proposed SDAFormer is shown in Figure 1, which consists of two stages: building localization (Stage 1) and damage classification (Stage 2). In Stage 1, a branch of SDAFormer, the segmentation network based on Swin Transformer, is used for building localization. This branch uses only the pre-disaster images as input and generates the building location masks. In Stage 2, both the pre-disaster and post-disaster images are fed into the siamese SDAFormer branches, respectively. Damage assessment is performed in Stage 2. Each branch of SDAFormer constitutes a Swin Transformer-based U-Net-like network. The weights are initialized from the trained weights of Stage 1 and shared between the two branches to better exploit the correlation between the pre- and post-disaster images.

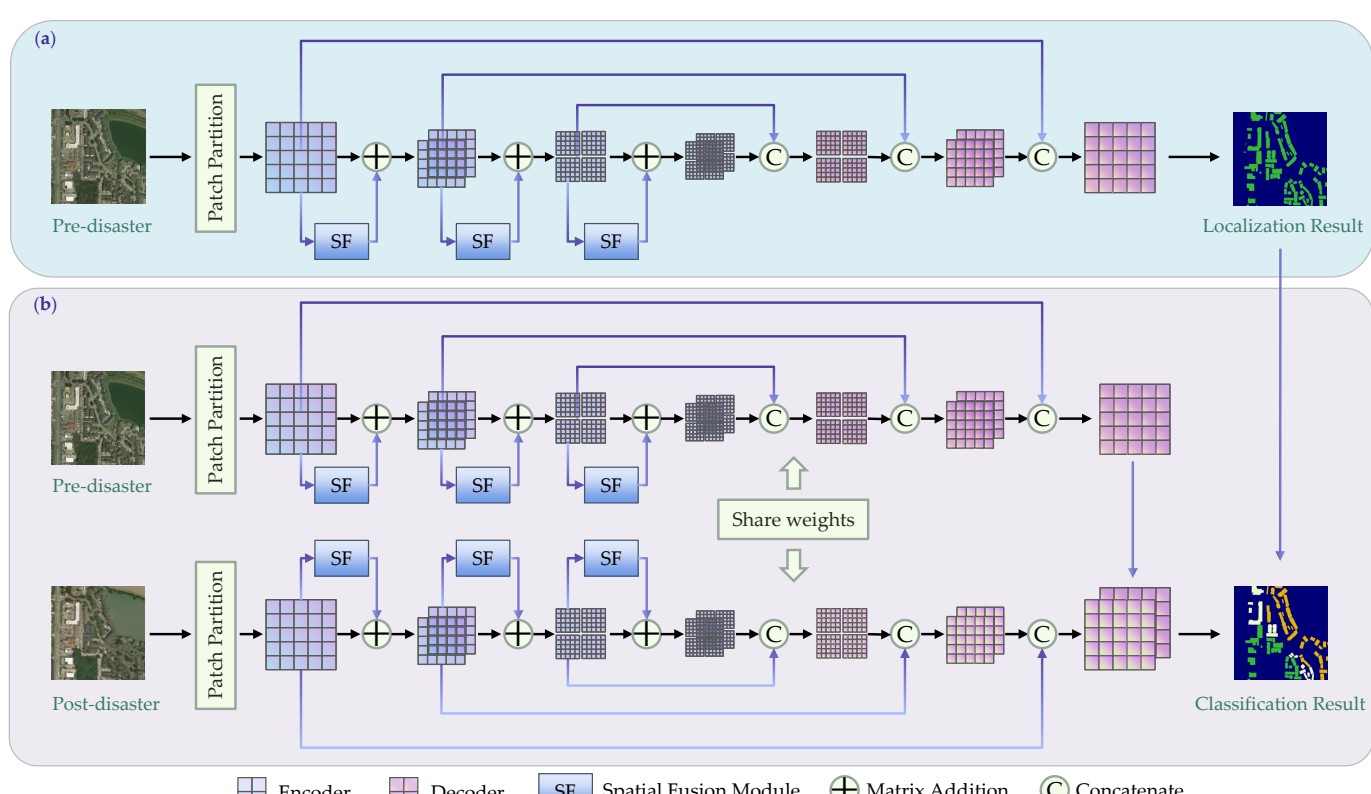

**Figure 1.** An overview of the SDAFormer framework. It is composed of two stages: (**a**) Stage 1: building localization, (**b**) Stage 2: damage classification.

## 2.2. Two-Stage Framework

Both tasks of change detection and damage assessment are based on the detection of the changed features of the image pairs, where the siamese structure is widely used for the network. However, compared with the goal of change detection, which is to find the regions of change between two temporal images, the building damage assessment task has some key differences. To fit the building damage assessment task, we use a two-stage framework and split the task into building localization and damage classification instead of directly applying the siamese network. The reasons are as follows.

- Firstly, instead of simply detecting changes, the damage level of each building also needs to be classified from the changes in the building damage assessment task. Therefore, locating buildings from the pre-disaster images provides more precise localization results and guides the damage classification in Stage 2.
- Secondly, the intact buildings are also required to be detected. In other words, the unchanged building objects need to be located and classified as the level of no damage, which is not required in the change detection task.
- Thirdly, a two-stage framework helps resolve the offset noise between two temporal images. In the building localization, only the pre-disaster image is used to ignore the offset noises from the post-disaster image. Moreover, the model trained in Stage 1 is transformed into the siamese network for classification, and the weights are shared. This strategy allows the network to process the pre-disaster and post-disaster images using the same approach, which helps overcome the offset noises.

### 2.2.1. Stage 1: Building Localization

The building segmentation is performed in Stage 1 to locate the buildings. As shown in Figure 1a, in Stage 1, only the pre-disaster images are input to train the network for the localization of buildings. Compared with the post-disaster images, the building contours in the pre-disaster images are more distinct to perform building segmentation. A single-branch U-Net-like network is used, which is shown in Figure 2. The Swin Transformer block [41] is the basic unit of the network. Patch partition is conducted for the encoder, in which the inputs are partitioned into non-overlapping patches of size $4 \times 4$ pixels to generate sequence embeddings. Then, a linear embedding layer is introduced to flatten and project patches into dimension $C$. These transformed patch tokens are put into the encoder for feature extraction, which consists of patch merging layers and the Swin Transformer blocks embedded with the SF module. The output feature map of the encoder is fed into a bottleneck consisting of two consecutive standard Swin Transformers, in which the number of channels and resolution of the feature map are kept constant. The decoder consists of standard Swin Transformer blocks and patch expanding layers, in which the patch expanding layer is responsible for up-sampling. By skip connections, the multi-scale features from the encoder can be fused with the extracted features to compensate for the loss of spatial information due to down-sampling. The network outputs a binary segmentation mask to locate the building objects before the disaster, and the results are used to guide the damage classification in the next stage.

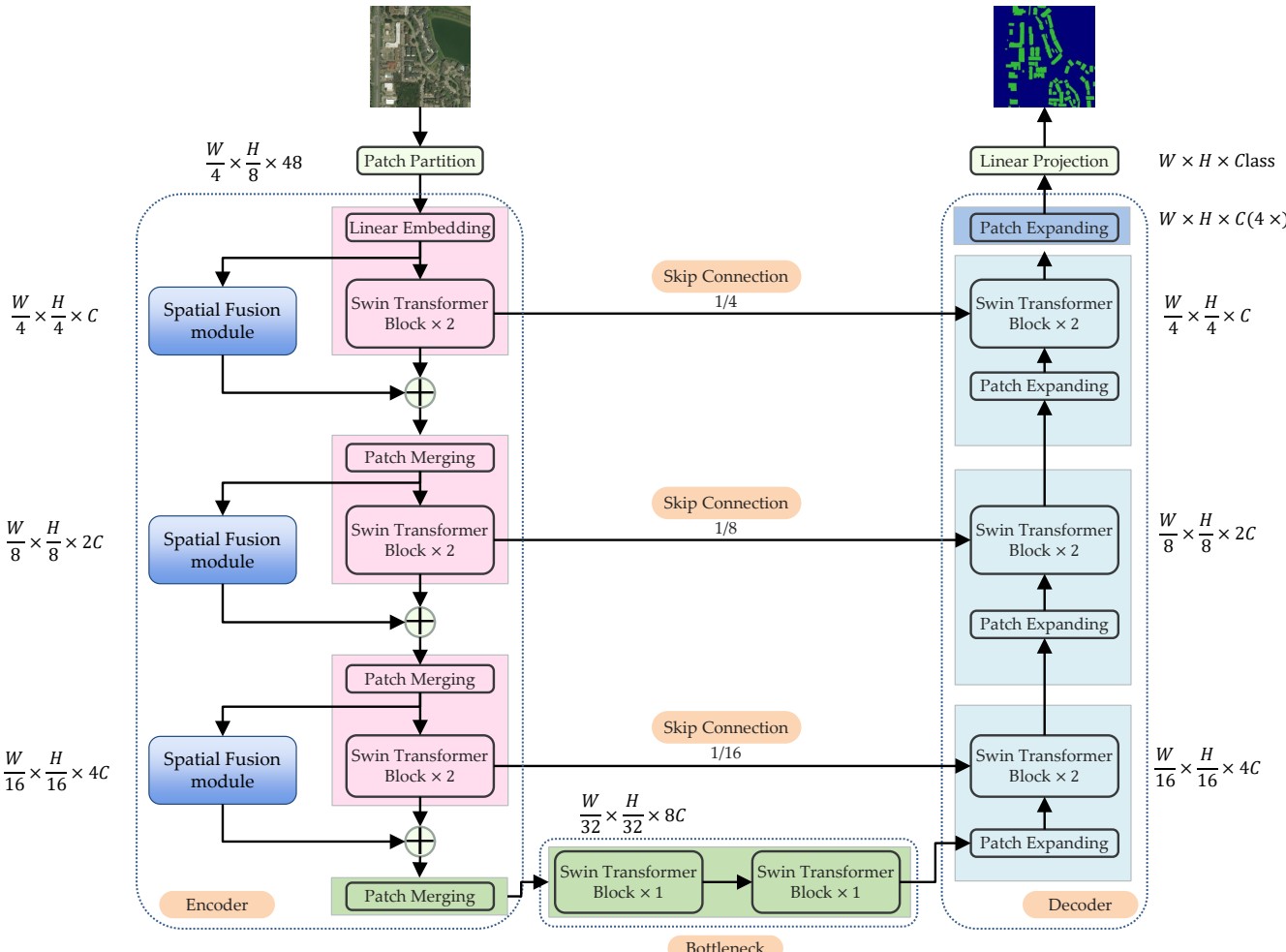

**Figure 2.** The architecture of U-Net-like network based on Swin Transformer block, which is composed of encoder, bottleneck, decoder and skip connections. Spatial fusion modules are embedded in the encoder.

### 2.2.2. Stage 2: Damage Classification

In Stage 2, the same backbone network as in Stage 1 is used, as shown in Figure 1b. In this stage, the weights from Stage 1 are used here for weight initialization. The pre-disaster and post-disaster images are sent into the two-branch network separately. The feature maps generated by the two branches are concatenated and put into a convolution layer of $1 \times 1$ so that the location information of the pre-disaster images can guide the damage classification of post-disaster images. The generated output is then used for the final classification of damage levels.

### 2.3. Swin Transformer Block

The traditional transformer block includes an MSA (Multi-head Self-Attention) module, MLP (MultiLayer Perceptron) and LN (Layer Normalization) layer. The output $s^l$ of layer $l$ is represented as [35]

$$\hat{z}^l = \text{MSA}\left(\text{LN}(z^{l-1})\right) + z^{l-1}, \tag{1}$$

$$z^l = \text{MLP}\left(\text{LN}(\hat{z}^l)\right) + \hat{z}^l \tag{2}$$

where $z^l$ represents the output of the Transformer block.

The computational costs of Transformer blocks grow quadratically as the number of tokens increase, which limits its application, especially on high-resolution remote sensing images. Therefore, Swin Transformer [41] proposes the shifted window mechanism. The Swin Transformer blocks consist of two sequential units, i.e., Window-based Multi-head Self-Attention (W-MSA) and Shifted Window-based Multi-head Self-Attention (SW-MSA), which is shown in Figure 3.

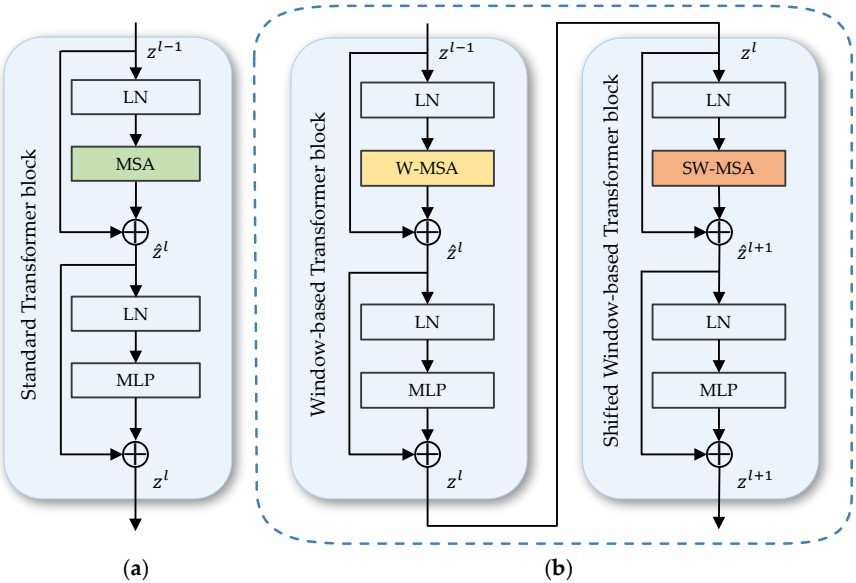

**Figure 3.** (**a**) The structure of a standard Transformer block [35]. (**b**) Two consecutive Swin Transformer blocks [41], which are called Window-based Transformer block and Shifted Window-based Transformer block, respectively.

In the W-MSA, the image is divided into non-overlapping windows of a certain size of $N$, and each window contains $N \times N$ patches. Self-attention is conducted within local windows, relieving the problem of high computing costs in MSA. To keep the correlation across windows, the SW-MSA introduces shifted windows partitioning, which brings information communication between adjacent non-overlapping windows of the previous layer.

Self-attention is conducted within windows, each of which covers $N \times N$ patches. $N$ is set to 8 to fit in with the resolution in this experiment. These two Swin Transformer blocks are renamed as Window-Transformer (W-Trans) block and Shifted Window-Transformer (SW-Trans) block. Two successive W-Trans and SW-Trans can be formulated as [41]

$$\hat{z}^l = \text{W-MSA}\left(\text{LN}(z^{l-1})\right) + z^{l-1}, \tag{3}$$

$$z^l = \text{MLP}\left(\text{LN}(\hat{z}^l)\right) + \hat{z}^l, \tag{4}$$

$$\hat{z}^{l+1} = \text{SW-MSA}\left(\text{LN}(z^L)\right) + z^l, \tag{5}$$

$$z^{l+1} = \text{MLP}\left(\text{LN}(\hat{z}^{l+1})\right) + \hat{z}^{l+1} \tag{6}$$

where $z^l$ represents the output of the W-Trans block, and $z^{l+1}$ represents the output of the SW-Trans block. Self-attention [49,50] is calculated as

$$\text{Attention}(Q, K, V) = \text{SoftMax}\left(\frac{QK^T}{\sqrt{d}} + B\right)V, \tag{7}$$

where $Q, K, V \in \mathbb{R}^{N^2 \times d}$ represent the query, key and value matrices, respectively. $N^2$ denotes the number of patches of a window. $B$ and $d$, respectively, represent the relative position bias and the dimension of the key or query. $T$ denotes matrix transposition.

### 2.4. Encoder

As shown on the left part of Figure 2, the patch partition divides the image into the non-overlapping patches. After the patch partition, the input image is reshaped to be $\frac{W}{4} \times \frac{H}{4} \times 48$, where $W$ and $H$ refer to the width and height of the original input image. The number of image patches is $\frac{W}{4} \times \frac{H}{4}$. Then, the patches are sent into the linear embedding layer. The linear embedding layer produces tokenized input with the dimension of $\frac{W}{4} \times \frac{H}{4} \times C$. A consecutive structure of dual Swin Transformer blocks embedded with SF module is developed, which is shown in Figure 4. Following three groups of such a structure and the patch merging layer, the encoder outputs the feature of $\frac{W}{32} \times \frac{H}{32} \times 8C$. In the process, Swin Transformer blocks are responsible for the representation learning without changing the size of feature dimension and resolution. Then, the patch merging layer divides the input patches into 4 equal parts and concatenates them together. The feature size is downsampled by $2\times$ after the procedure. To reduce the dimension of concatenated features from $4\times$ to $2\times$, a linear layer is utilized in the patch merging layer.

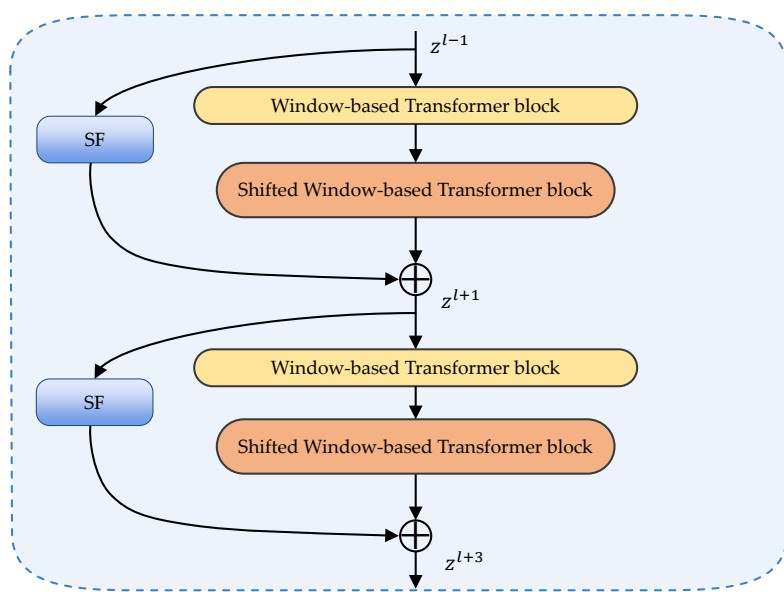

**Figure 4.** Swin Transformer blocks with the spatial fusion (SF) module.

### 2.5. Bottleneck

In the bottleneck, the resolution and dimension of the feature remain unchanged. Two Swin Transformer blocks are used to replace the CNN-based bottleneck block for better deep feature representation [51].

An MSA layer can be used as a convolution layer in the bottleneck. Consider an MSA layer consisting of $N_h = K^2$ heads of dimension $D_h$ and output dimension $D_{out}$. Let $f : [N_h] \to \Delta_K$ be a mapping of heads onto shifts and $\Delta_K$ contain all possible shifts in the convolution. Suppose that the following holds for every head $h$:

$$\text{softmax}\left(A_q^{(h)}\right)_k = \begin{cases} 1 & \text{if } f(h) = q - k \\ 0 & \text{otherwise} \end{cases} \tag{8}$$

where $q, k$ stand for query and key pixels. $A$ stands for the attention scores and the softmax outputs attention probabilities. Then, for any convlution layer (Conv) with $D_{out}$ output dimension, there are $\left\{ W_{\text{val}}^{(h)} \right\}_{h \in [N_h]}$ such that $\text{MSA}(X) = \text{Conv}(X)$ for every $X \in \mathbb{R}^{W \times H \times D_{\text{in}}}$. For one output pixel of MSA:

$$\text{MSA}(\boldsymbol{X})_{q,:} = \sum_{h \in [N_h]} \left( \sum_k \text{softmax}\left(\mathbf{A}_{q,:}^{(h)}\right)_k \mathbf{X}_k \right) \boldsymbol{W}^{(h)} + \boldsymbol{b}_{\text{out}} \qquad (9)$$

where $\boldsymbol{W}$ is the weight tensor and $\boldsymbol{b}$ is the bias vector. For the $h$-th head, the probability of attention is 1 when $\boldsymbol{k} = \boldsymbol{q} - \boldsymbol{f}(h)$ and 0 otherwise. The output is then equal to:

$$\text{MSA}(\boldsymbol{X})_q = \sum_{h \in [N_h]} \mathbf{X}_{q-f(h),:} \boldsymbol{W}^{(h)} + \boldsymbol{b}_{\text{out}} \qquad (10)$$

When $K = \sqrt{N_h}$, the above is equivalent to a convolution layer. Therefore, the bottleneck using Swin Transformer blocks performs the same function as a convolutional bottleneck.

### 2.6. Decoder

Symmetric with the encoder, the decoder is developed based on regular Swin Transformer blocks and patch expanding layers [42] to up-sample the feature map to the original resolution, as shown on the right part of Figure 2. Contrary to the patch merging layer in the encoder, the patch expanding layer up-samples the extracted feature by 2×. Specifically, given the dimension of the input is $2C_1$, the patch expanding layer uses a linear layer to double the feature dimension to $4C_1$. Then, a rearrange operation is conducted to expand the resolution of the input features by 2×, and the dimension of the feature is reduced to one-quarter of the ascended dimension to be $C_1$.

With skip connections, the up-sampled features in the decoder are fused with the features from the encoder. Shallow features are concatenated with deep features to reduce the loss of the spatial information during down-sampling, and a linear layer keeps the dimension unchanged. The process of skip connection can be formulated as:

$$y^{l+1} = F(y^l \copyright x^l) \qquad (11)$$

where $x^l$ and $y^l$ refer to the feature from the encoder and the decoder, respectively, while $y^{l+1}$ refers to the input of rearrange operation. $\copyright$ represents the channel-level concatenation. $F(\cdot)$ represents the linear layer.

### 2.7. Spatial Fusion Module

In Swin Transformer, the computation of self-attention is limited in each local non-overlapping window, resulting in linear computational cost in relation to image size and reduced memory overhead. However, this mechanism cuts down the ability of global modeling compared to Transformer [4]. Moreover, additional spatial knowledge helps eliminate the blurred contours of the post-disaster building. Therefore, the Spatial Fusion (SF) module is proposed to allow for more spatial knowledge learned by the encoder and boost the information exchange over the W-Trans block and SW-Trans block.

By adding the weighted feature to the output of the W-Trans block, the SF module emphasizes spatial information from neighboring pixels of the same class label as the central pixel, and it suppresses pixels of different class labels. The structure of the SF module is shown in Figure 5. In phase $n$, let the input feature of W-Trans block $z^{l-1} \in \mathbb{R}^{(h \times w) \times c_1}$ reshape into $s \in \mathbb{R}^{h \times w \times c_1}$, where $c_1 = 2^{n-1}C$, $w = \frac{W}{2^{n+1}}$ and $h = \frac{H}{2^{n+1}}$. The $s$ is sent into a $3 \times 3$ dilated convolution layer of dilation rate 2.

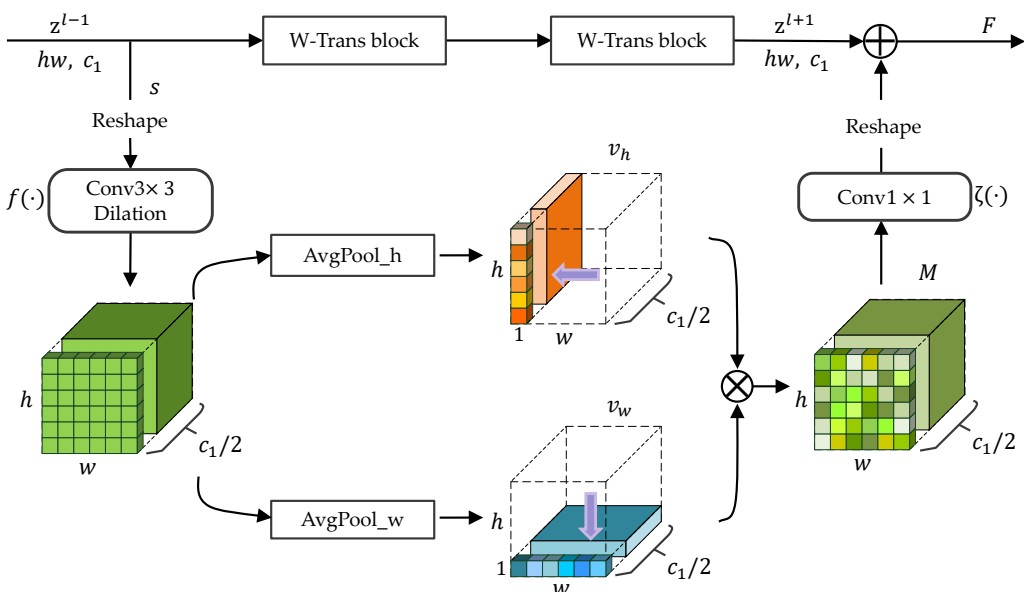

**Figure 5.** The structure of SF module.

A simple convolution layer can be formed as:

$$\hat{s}(x,y) = \sum_{d'=1}^{c_1} \left( \sum_{-L \leq i,j \leq L} k_d(i,j,d') \cdot F_{d'}(x-i,y-j) \right) \tag{12}$$

where $x$, $y$ are integers for which $0 \leq x \leq w$, $0 \leq x \leq h$, and $1 \leq d' \leq c_1$. $K_d$ stands for coefficients for all pairs $(i,j,d')$ for which $|i|, |j| \leq L$. $F_{d'}$ is the input and $\hat{s}$ is the output of the layer. It can be inferred that $\hat{s}$ corresponding to the point $(x,y)$ is determined by $F_{d'}(x-i,y-j)$ at points $(x-i,y-j)$ corresponding to $|i|, |j| \leq L$.

Compared with convolution, dilated convolution makes a larger receptive field. A dilated convolution layer is formed as:

$$\hat{s}(x,y) = \sum_{d'=1}^{c_1} \left( \sum_{-L \leq i,j \leq L} k_d(i,j,d') \cdot F_{d'}(x-\ell \cdot i,y-\ell \cdot j) \right) \tag{13}$$

where $\ell$ refers to the dilation rate. In this case, convolution is performed on $(x-\ell \cdot i, y-\ell \cdot j)$ instead of $(x-i,y-j)$. Dilated convolution is conducted within more distant pixels. This approach expands the receptive field and thereby enhances the perception of global information.

Then, the channel number is reduced to $c_1/2$, and the feature passes through the global average pooling layer in the horizontal and vertical directions, respectively, to get the feature map statistics of spatial information. The formula of the above two directions is represented as:

$$v_{w_j}^i = \frac{1}{h} \sum_{j=0}^{h-1} \hat{s}^i(j,k), \tag{14}$$

$$v_{h_k}^i = \frac{1}{w} \sum_{j=0}^{w-1} \hat{s}^i(j,k), \tag{15}$$

where $i$, $j$, and $k$ are the indexes of the channel and the horizontal and vertical directions, respectively, and $0 \leq k \leq h$, $0 \leq j \leq w$, $0 \leq i < c_1/2$. The tensor $v_w$ contains the attention weights in the horizontal direction and $v_h$ in the vertical direction. Then, the $v_w$ and $v_h$ obtained from (14) and (15) multiply to obtain the attention feature map on spatial, $M \in R^{h \times w \times \frac{c_1}{2}}$, following a convolution layer to double the dimension. The final feature

map $F$ output from SF is generated by adding $M$ and $s^{l+1}$ yielded from the SW-Trans block. The feature $F \in R^{h \times w \times c_1}$ is formulated as

$$M = v_w \otimes v_h \tag{16}$$

$$F = z^{l+1} \oplus \zeta(Ge(M)) \tag{17}$$

where $\oplus$ refers to element-level addition, and $\otimes$ refers to matrix multiplication. $\zeta(\cdot)$ and $Ge(\cdot)$ stand for the $1 \times 1$ convolutional layer and GELU activation function, respectively.

## 3. Experiment Results

### 3.1. Experiment Data

3.1.1. xBD Dataset

In our study, the xBD dataset [48] is used to validate the performance of the proposed method. The xBD dataset is a large-scale public building segmentation and damage assessment dataset with high-quality building annotations from high-resolution satellite images before and after 19 different natural disasters (e.g., earthquakes, volcanic eruptions, hurricanes, and floods). It is sourced from the Maxar/DigitalGlobe Open Data Program [52], where high-resolution images from many disparate regions of the worlds are available. The dataset consists of pairs of pre-disaster and post-disaster $1024 \times 1024$ satellite images. The images are below the 0.8 m ground sample distance (GSD) mark. The split of train, validation, and test sets is shown in Table 1.

**Table 1.** Size of xBD dataset split.

| Split | Image Number |
|:---:|:---:|
| Train | 16,470 |
| Validation | 1833 |
| Test | 1866 |

The dataset provides 4-level damage labels, including no damage, minor damage, major damage and destroyed. The number of damage annotations of each level is shown in Table 2. It should be noted that the distribution of each damage level is imbalanced.

**Table 2.** The distribution of damage level annotations.

| | No Damage | Minor | Major | Destroyed |
|:---:|:---:|:---:|:---:|:---:|
| Number | 313,033 | 38,680 | 29,904 | 31,560 |
| Percentage | 76.04% | 8.98% | 7.29% | 7.69% |

3.1.2. Instance Data

Four individual disaster events are used to verify the robustness and transferability of the proposed method. Two of them are the tornadoes in USA, and the other two are Typhoon Yutu in the northern Mariana Islands. The detailed information of these disasters is shown in Table 3.

**Table 3.** Detailed information of the two disasters.

| Disaster | Location | Date |
|:---:|:---:|:---:|
| Tornado in Arkansas | Monette, AR, USA | 20 December 2021 |
| Tornado in Kentucky | Mayfield, KY, USA | 28 January 2017 |
| Typhoon Yutu | Saipan Island, Northern Mariana Islands | 12 October 2018 |
| Typhoon Yutu | Tinian Island, Northern Mariana Islands | 12 October 2018 |

### 3.2. Implementation Details

The proposed method is implemented using Pytorch 1.10. The experimental environment is on a computer with an Intel Core i7-10700 CPU and a NVIDIA RTX-3090 GPU. Simple data argumentation is used to enhance the diversity of the data, including rotation and flip. AdamW is used as the optimization algorithm for backpropagation. The learning rate for Stage 1 (building localization) is 0.00015 and for Stage 2 (damage classification) is 0.0002. The number of epochs for Stage 1 is 120 and for Stage 2 is 20. The pre-trained weights with ImageNet are used for initialization.

### 3.3. Loss Function

We adopt binary cross-entropy loss for building localization loss $L_{loc}$, which is defined as

$$L_{loc} = -[y_{loc} \log P_{loc} + (1 - y_{loc}) \log(1 - p_{loc})] \tag{18}$$

where $P_{loc}$ and $y_{loc}$ are the probability of building location and the reference label. The damage classification outputs a mask of five channels, including one channel of localization and four of damage levels. In order to alleviate the imbalance of samples on damage levels, a weighted mixed loss function which consists of focal loss and dice loss is used for damage classification loss $L_{cls}$, which is formulated as:

$$L_{cls} = \sum_{n=0}^{4} w_n \times [c_1 \times \text{Focal}_n(m_p, m_t) + c_2 \times \text{Dice}_n(m_p, m_t)] \tag{19}$$

where $m_p$ and $m_t$ are the predicted mask and true mask for channel $n$, respectively. $c_1$ and $c_2$ are the weights for focal loss and dice loss, respectively. $w_n$ is the weight for channel $n$. Larger weights are set for minor damaged and major damaged, which are uncommon classes ($c = 2, 3$). Accordingly, a smaller weight is set for localization weight ($c = 0$).

### 3.4. Performance Evaluation Metrics

In segmentation tasks, precision and recall are important accuracy indicators. In most cases, it is difficult to evaluate performance well using only one of them. The F1 score represents the balance between precision and recall and can better reflect the overall performance of the model, especially in the case of unbalanced samples. TP (true-positive) represents the number of pixels that are predicted as the right categories. FP (false-positive) denotes the number of pixels from other categories that are incorrectly predicted as this category. FN (false-negative) indicates the number of pixels belonging to this category that are incorrectly classified. In this paper, the XView2 Challenge metric [48] is used to evaluate the results. The F1 for weighted mean of the building segmentation ($F1_{loc}$) and the F1 for harmonic average of class damage classification ($F1_{cls}$) are applied. $F1_{C_i}$ refers to the F1 score of each damage class and $C_i$ represents the $i$-th damage class, where $C_1$ to $C_4$ denote no damage, minor, major, and destroyed, respectively. $F1_{loc}$ and $F1_{cls}$ are defined as

$$F1_{loc} = \frac{2TP}{2TP + FP + FN}, \tag{20}$$

$$F1_{cls} = \frac{n}{\sum_{i=1}^{n} \frac{1}{F1_{C_i}}}, \tag{21}$$

The final score [48] of overall evaluation comprehensively reflects the building segmentation and damage classification performance, which is formed based on $F1_{loc}$ and $F1_{cls}$.

$$score = 0.3 \times F1_{loc} + 0.7 \times F1_{cls} \tag{22}$$

### 3.5. Comparisons with Other Models on xBD Dataset

To verify the effectiveness of the proposed method in this paper, we compare it with some existing CNN-based methods, including

- The Weber's method [53] utilizes Mask R-CNN with FPN structure and parallel architecture for both building segmentation and damage classification. It concatenates pre-disaster and post-disaster features after feature extraction with ResNet-50. Then, the fused feature map is fed into the segmentation head for damage assessment. A novel loss function is designed to weight the mistakes on levels inversely proportional to their occurrence in the xBD dataset.
- In RescueNet [54], a dilated ResNet-50 is used for the backbone of the U-Net. To utilize the differences between pre-disaster and post-disaster images, both images are fed into the network for building segmentation. Only the post-disaster image is used in the task of damage classification. Different loss functions are applied to the two tasks separately. Specifically, the Binary Cross-Entropy loss is used for building segmentation, while the foreground-only selective Categorical Cross-Entropy loss is used for damage classification. A dual-head framework is developed, which contains a segmentation head and a change detection head.
- The approaches of the top two results from the XView2 Challenge are employed for evaluation, including XView2 1st [55] and XView2 2nd [56]. The XView2 1st builds a multi-model ensemble for better performance. XView2 2nd simultaneously applies DPN-92 and DenseNet-161 to U-Net for damage assessment. Both methods use various techniques including data argumentation and multiple test strategies.

Table 4 shows the damage assessment results of different methods on the xBD dataset. In the overall task, our framework performs better and reaches the score of 80.2%. In comparison with the highest overall score of XView2 1st, our method obtains a 1.5% boost. The $F1_{loc}$ is improved compared with MaskRCNN and RescueNet. The $F1_{C_i}$ scores also reach their highest, except that the $F1_{C_1}$ is slightly lower than the method of XView2 2nd, which may be due to the dual-model strategy of XView2 2nd.

**Table 4.** Performance comparison of the proposed method with existing CNN-based methods. ($F1_{C_1}$: $F1$ score of no damage; $F1_{C_2}$: $F1$ score of minor damage; $F1_{C_3}$: $F1$ score of major damage; $F1_{C_4}$: $F1$ score of destroyed).

| Methods | Score (%) | $F1_{loc}$ (%) | $F1_{cls}$ (%) | $F1_{C_1}$ (%) | $F1_{C_2}$ (%) | $F1_{C_3}$ (%) | $F1_{C_4}$ (%) |
|---|---|---|---|---|---|---|---|
| MaskRCNN [53] | 74.1 | 83.6 | 70.0 | 90.6 | 49.3 | 72.2 | 83.7 |
| RescueNet [54] | 77.0 | 84.0 | 74.0 | 88.3 | 56.3 | 77.1 | 80.8 |
| XView2 1st | 78.7 | 86.1 | 75.5 | 91.9 | 57.2 | 77.0 | 86.3 |
| XView2 2nd | 76.8 | 84.0 | 73.7 | **92.8** | 53.8 | 75.2 | 85.9 |
| SDAFormer | **80.2** | **86.1** | **77.6** | 92.5 | **61.4** | **77.5** | **86.8** |

Figure 6 visualizes the damage assessment results of each method. For RescueNet, more errors appear in the damage classification, especially on the minor damage level. For Mask R-CNN, it can be seen that the model outputs more segmentation mistakes, and more mistakes appear on the edges of damaged areas. It can be seen that our proposed model obtains more accurate damage level prediction with smoother boundaries. Overall, the proposed SDAFormer performs best with fewer assessment mistakes.

Figure 7 visualizes the results of the building segmentation, which is Stage 1 of the proposed framework. The results of Figure 7c,d mistakenly detect the area of cropland as buildings and cannot provide accurate contours of the buildings in the pre-disaster image. It can be seen that our framework achieves more precise segmentation results in comparison with the other methods. In the results of our method, the contours are clear enough for the siamese network to locate the buildings. Thus, the output of Stage 1 is adequate for locating buildings in Stage 2 where damage classification is to be performed.

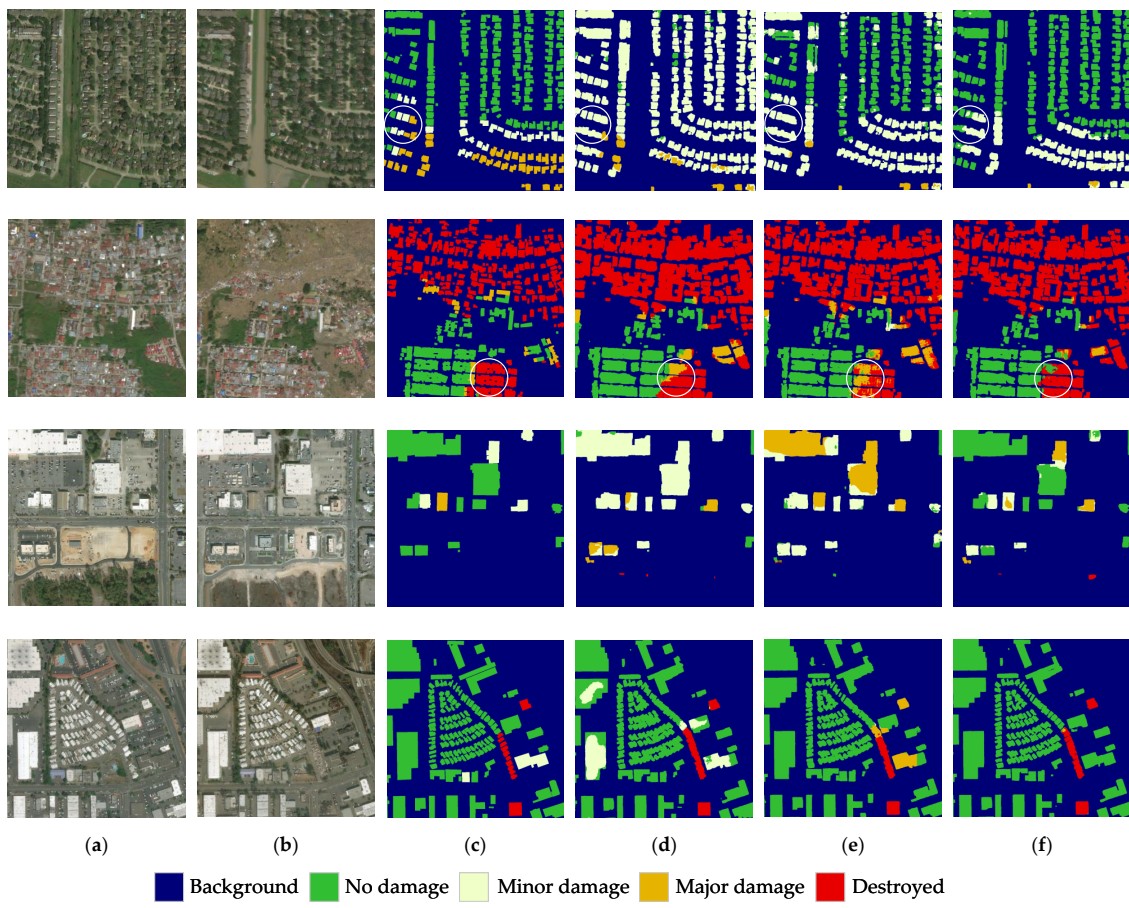

(**a**)　　　(**b**)　　　(**c**)　　　(**d**)　　　(**e**)　　　(**f**)

■ Background　■ No damage　■ Minor damage　■ Major damage　■ Destroyed

**Figure 6.** Building damage assessment results. (**a**,**b**) respectively show pre- and post-disaster images; (**c**) shows ground truth; (**d**,**e**) are the results of RescueNet and MaskRCNN, respectively; (**f**) is the prediction of our proposed framework.

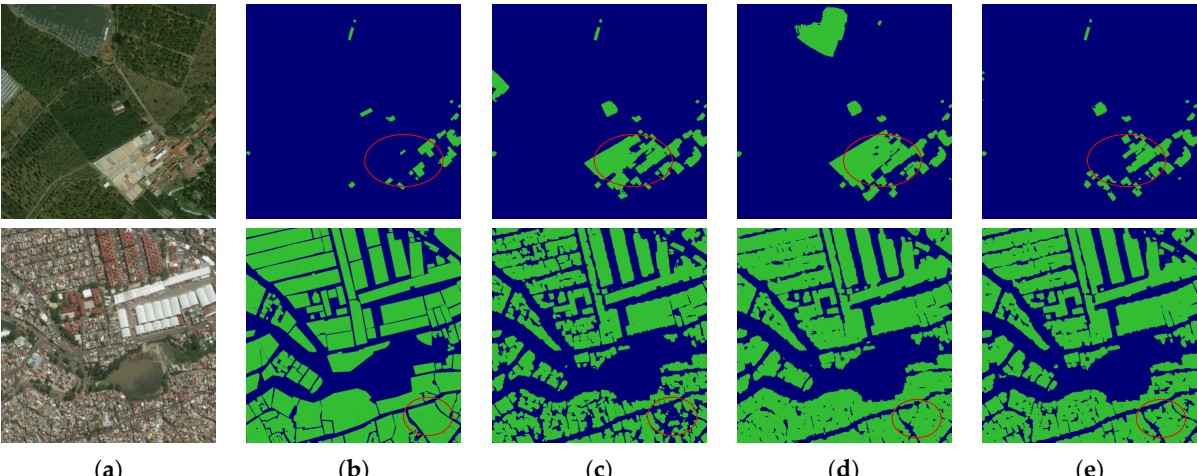

(**a**)　　　　　(**b**)　　　　　(**c**)　　　　　(**d**)　　　　　(**e**)

**Figure 7.** Visual examples of segmentation results in Stage 1. (**a**) shows pre-disaster images; (**b**) shows ground truth; (**c**,**d**) are the results of RescueNet and MaskRCNN, respectively; (**e**) is the segmentation result of our proposed framework.

*3.6. Ablation Study*

An ablation experiment is conducted to demonstrate the effectiveness of our proposed method. To investigate the effectiveness of the Transformer-based structure, a CNN-based network using the backbone of Res-50 without additional modules as the baseline is

implemented for comparison. To evaluate the contribution of the SF module, a Transformer-based network without the SF module is used. Table 5 shows that the scores of Transformer-based methods exceed the score of the Res-50-based baseline, which proves the effectiveness of the Transformer-based network.

**Table 5.** Ablation study of the proposed method ($F1_{C_1}$: $F1$ score of no damage; $F1_{C_2}$: $F1$ score of minor damage; $F1_{C_3}$: $F1$ score of major damage; $F1_{C_4}$: $F1$ score of destroyed).

| Methods | Score (%) | $F1_{loc}$ (%) | $F1_{cls}$ (%) | $F1_{C_1}$ (%) | $F1_{C_2}$ (%) | $F1_{C_3}$ (%) | $F1_{C_4}$ (%) |
|---|---|---|---|---|---|---|---|
| Baseline (Res-50) | 78.5 | 85.9 | 75.4 | 92.2 | 57.0 | 76.3 | 86.3 |
| SDAFormer (without SF) | 79.5 | 86.1 | 76.7 | **92.6** | 59.6 | 76.9 | 86.5 |
| SDAFormer (with SF) | **80.2** | **86.1** | **77.6** | 92.5 | **61.4** | **77.5** | **86.8** |

Furthermore, SDAFormer achieves an impressive improvement with the SF module. It can be seen that the $F1$ score of damage classification ($F1_{cls}$) is improved to 77.6%. The result shows that the SF module has little influence on the localization accuracy, but obvious performance gain is achieved on the classification accuracy. The score of minor damage class ($F1_{C_2}$) obtains a better improvement (from 57.0% to 61.4%), and it shows that the spatial attention mechanism helps enhance the performance on the damage level, which is difficult to recognize.

To further evaluate the influence of the Transformer structure and SF module, three groups of sample images in the test set are picked out for comparison, as shown in Figures 8–10, respectively.

Figure 8 illustrates the performance on the detection of undamaged and minor damaged buildings. It can be seen that SDAFormer has more prediction mistakes on minor damage. There are several reasons for this. Firstly, the training set is imbalanced on the damage level annotations and heavily biased toward the level of no damage. Secondly, the high visual similarity between no damage and minor damage leads to the misclassification of these two classes. Moreover, as shown in Figure 8a,b, the imaging angles of the pre-disaster and post-disaster images are different, which leads to an incomplete overlap of building locations in the two images.

Figure 9 illustrates the performance on the detection of undamaged and major damaged buildings. It can be seen that the Transformer-based models perform well in detecting undamaged buildings. However, due to the complex damage distribution of the local buildings, wrong judgments are made for the major damaged buildings. The baseline method achieves a lower prediction accuracy, where some undamaged buildings are incorrectly classified as major damaged buildings. In comparison with the result of the SDAFormer without the SF module, the SDAFormer with SF module outputs more accurate results on the major damage classification.

Figure 10 illustrates the performance on the detection of destroyed buildings. The assessment result shows that the output of all models is generally correct in terms of building localization. As for the damage assessment, the baseline output in Figure 10d has a few classification errors in the lower left corner of the image, and the quality of the assessment for small building objects is not satisfactory. The output of SDAFormer can indicate the damage degree of the disaster-affected area, but the results differ in the details. For example, the building group in the center of Figure 10e is regarded as major damage by SDAFormer without SF. In Figure 10f, the building group is regarded as undamaged except for the lower right corner. Comparing the pre-disaster and post-disaster buildings in Figure 10a,b, due to the tsunami, it can be recognized that a giant deviation of the buildings emerged after the disaster, but the structures of the buildings are still largely preserved. However, such buildings are taken as undamaged in the ground truth masks in xBD. The model with SF notices the structural connections of the deviant houses and shows a better performance in this position.

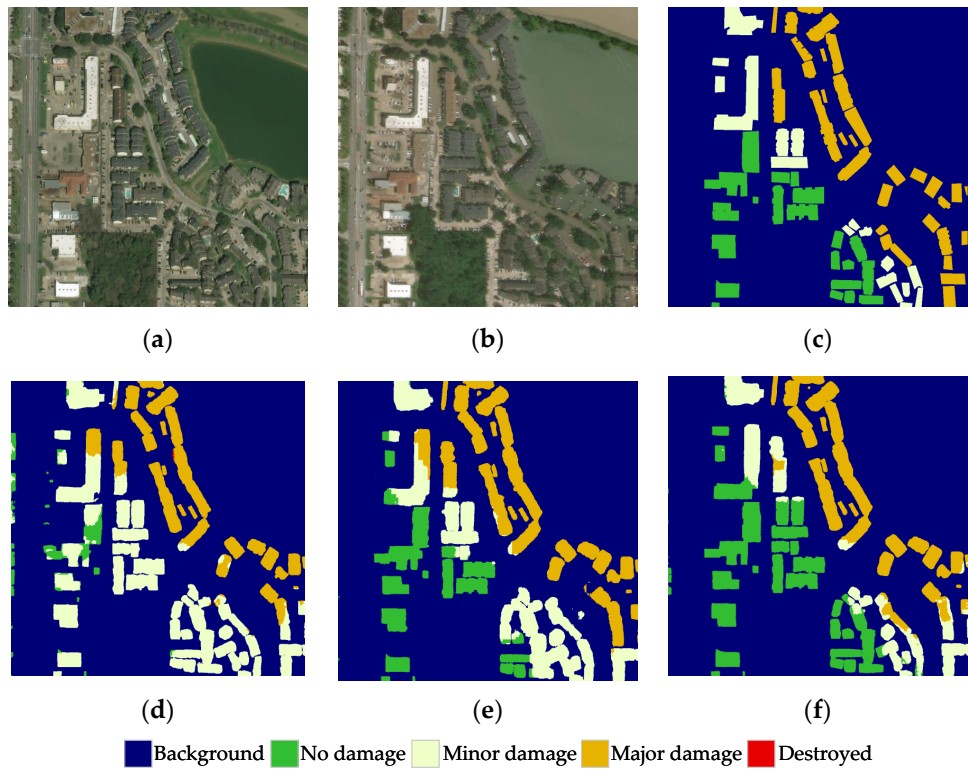

**Figure 8.** The visual comparison of prediction results on hurricane–Harveyan xBD dataset. (**a**) Pre-disaster; (**b**) post-disaster; (**c**) ground truth; (**d**) baseline; (**e**) SDAFormer (without SF); (**f**) SDAFormer (with SF).

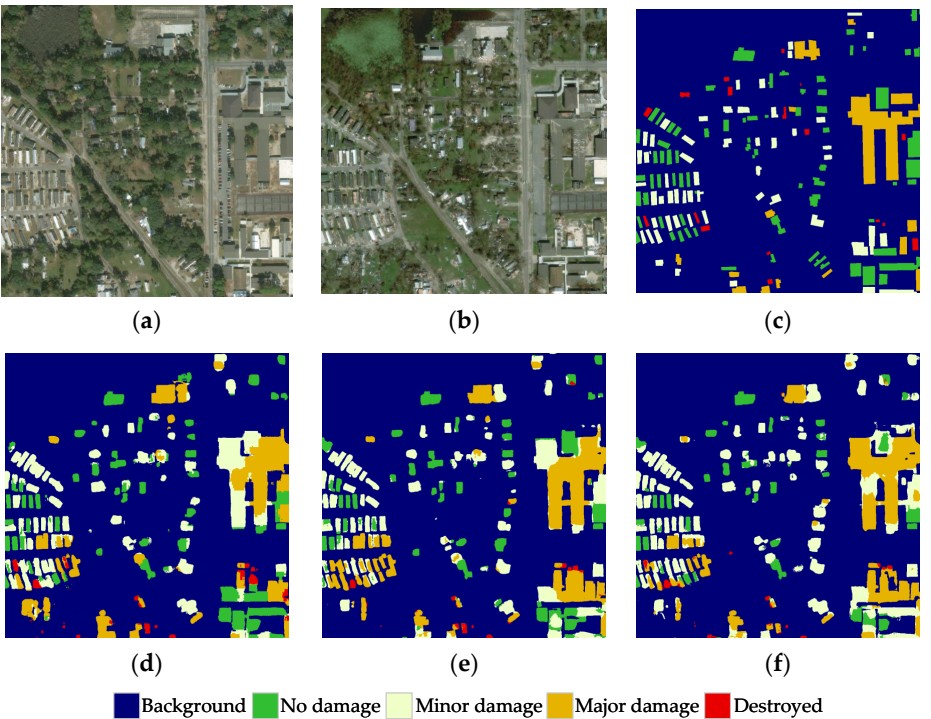

**Figure 9.** The visual comparison of prediction results on hurricane-Michael in xBD dataset. (**a**) Pre-disaster; (**b**) post-disaster; (**c**) ground truth; (**d**) baseline; (**e**) SDAFormer (without SF); (**f**) SDAFormer (with SF).

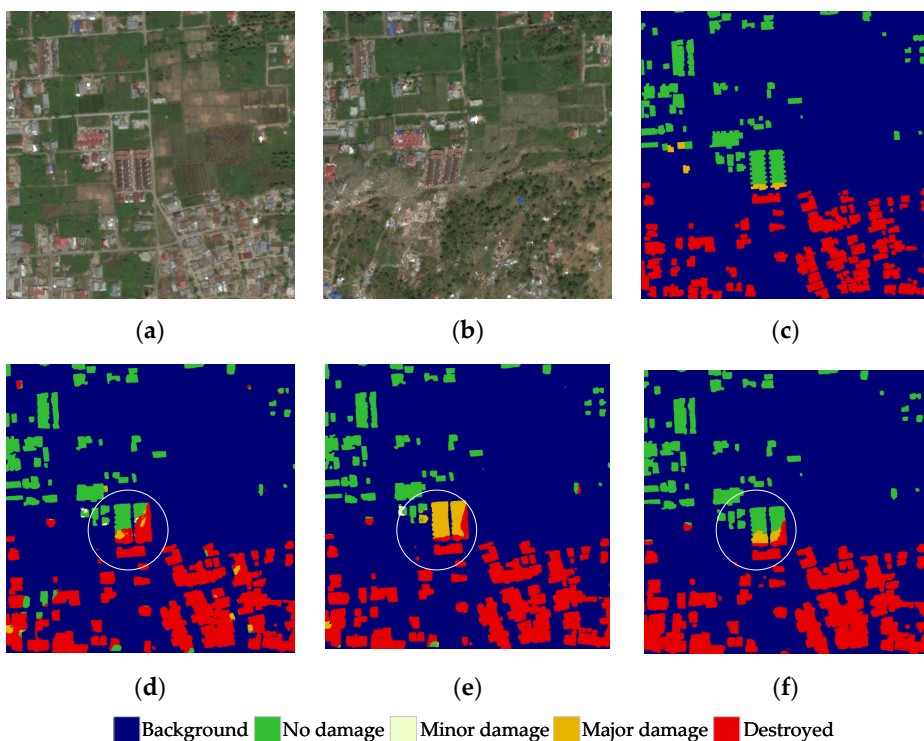

<div style="text-align:center">

■ Background ■ No damage □ Minor damage ■ Major damage ■ Destroyed

</div>

**Figure 10.** The visual comparison of prediction results on palu-tsunami in xBD dataset. (**a**) Predisaster; (**b**) post-disaster; (**c**) ground truth; (**d**) baseline; (**e**) SDAFormer (without SF); (**f**) SDAFormer (with SF).

### 3.7. Robustness and Transferability

Due to the difficulty in obtaining a building damage assessment dataset other than the xBD dataset, we selected four independent disaster events outside the xBD dataset to verify the transferability and robustness of the proposed method. The details of the events are listed in Table 3. For each instance, the pre-disaster and post-disaster images are fed into our framework. The results are shown in Figure 11.

In the cases of the tornadoes, the buildings are less dense. Due to the seriously damaged buildings, most of the buildings in the image are labeled as destroyed in our model, which is consistent with the damage situation. The results of RescueNet and MaskRCNN cannot accurately outline the buildings and therefore fail to predict the damage levels.

In the cases of typhoon Yutu, the results of our model shows that the majority of the buildings in the image are well detected and correctly located. Some multi-story buildings cast large areas of shadows on the upper right side, which limits the ability to locate buildings and causes some errors around these buildings. In the damage assessment stage, it can be seen that most of the single-story houses are correctly classified as reasonable damage levels from no damage to destroyed. However, for the multi-story buildings located in the right of the image, the majority of them are recognized as destroyed or major damage. It can be seen from the post-disaster image that the texture features of some of the roofs are changed. For some buildings which are detected as destroyed, the top floors are damaged but the structure of the buildings still remains. Moreover, the effect of side shooting distracts the assessment, which is common on high buildings. On the whole, the predicted results of our method are more precise than other compared methods.

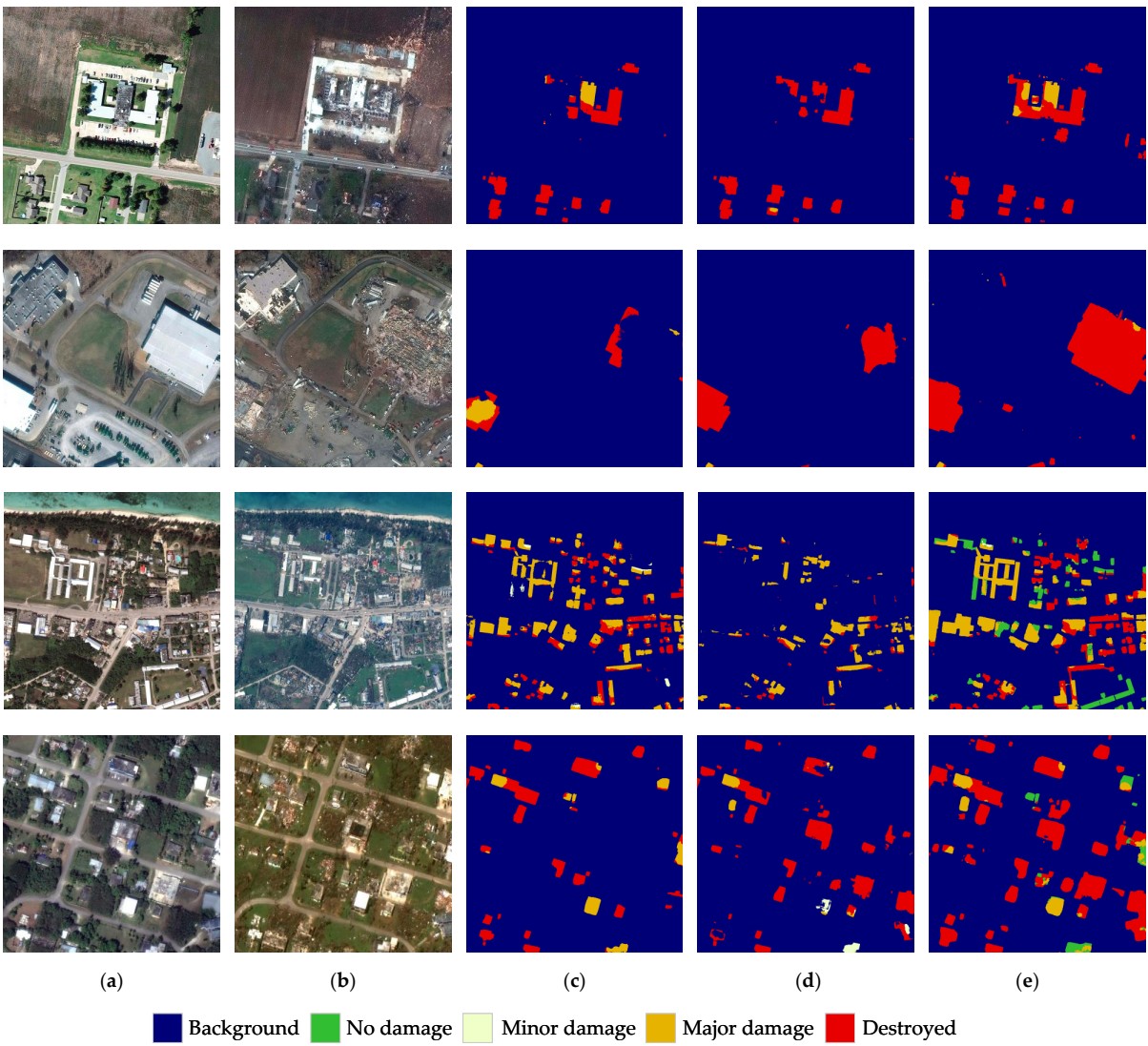

  (**a**)      (**b**)      (**c**)      (**d**)      (**e**)

■ Background ■ No damage ■ Minor damage ■ Major damage ■ Destroyed

**Figure 11.** Results of the independent disaster events. First row to fourth row are: tornado in Arkansas and in Kentucky, typhoon on the Saipan Island and on the Tinian Island. (**a**) Pre-disaster; (**b**) post-disaster; (**c**) RescueNet; (**d**) MaskRCNN; (**e**) our results.

## 4. Discussion

### 4.1. Findings and Implications

  Pairs of pre-disaster and post-disaster satellite images can reflect the building damage level in the disaster-affected areas in a timely and accurate manner. Therefore, we construct the two-stage SDAFormer framework. Meanwhile, Swin Transformer is introduced to form the framework. According to the analysis of experimental results, our framework has a higher overall score than existing CNN-based methods, which proves the effectiveness of our method. The proposed two-stage framework can consider the temporal and spatial relevance between pre- and post-disaster remote sensing images, which helps to improve the building segmentation and damage assessment.

  The application of different types of Transformers in the visual field has been a hot research topic in recent years, which can improve the scalability and performance of many tasks. Transformer can correlate key features in different channels and improve the ability to model the global relationships of the framework in building damage assessment. In our study, Swin Transformer is applied in the encoder, decoder, and bottleneck, which shows the universality of the Swin Transformer block in the building damage assessment field.

In our study, a spatial attention mechanism, also known as the spatial fusion module, is also introduced. For the model with the spatial attention mechanism, the score of building localization is kept constant. Meanwhile, the accuracy of damage classification is improved, especially in the minor and major damage classes. The spatial fusion module has little impact on the semantic segmentation performance of the module, which can use the surrounding texture of the buildings to support damage-level inference. The spatial fusion module is able to enhance the spatial context feature representation and compensate for the limitation of window mechanism in Swin Transformer. As a result, the ambiguity caused by blurred post-disaster buildings can be alleviated, improving the ability to distinguish between minor damage and major damage level.

According to Table 4, Weber's method (Mask R-CNN) has much lower $F1_{loc}$ and $F1_{cls}$ than SDAFormer. Unlike our proposed framework, in Weber's method, a single-branch decoder is applied after concatenating the extracted pre-disaster and post-disaster features. The single-branch decoder results in the inability to utilize the relevant features of the pre-disaster and post-disaster buildings. Therefore, buildings of different sizes are unable to be located properly, which affects the damage assessment result. Meanwhile, the CNN structure limits the global feature extraction capability and the overall understanding of the image. Regarding RescueNet, both pre-disaster and post-disaster images are simultaneously used for building segmentation, but only post-disaster images are used in the process of damage assessment. This strategy can provide additional information at different points in time to the segmentation task. However, since the structure of the buildings is damaged or destroyed by the disaster, the additional semantic information of the post-disaster images can lead to confusion of the network. Meanwhile, the damage classification task lacks the guidance on the localization of the original building, which weakens the ability to detect destroyed buildings because of the seriously damaged appearance.

### 4.2. Limitations

Although SDAFormer has shown superior performance in the experiments, it still has two limits.

- First, with the process of urbanization, it is important to assess urban disaster-affected areas. However, it is still difficult to comprehensively assess the damage of the multistory buildings for our proposed framework. The information of a building from the satellite image is limited to roofs, while damages on walls cannot be effectively detected. In some cases, these tall buildings are projected into irregular shapes due to the satellite imaging angle, and the sides of the buildings are shown in the satellite images, which may lead to ambiguity of the network. In addition, the shadows of the buildings may change with the satellite acquisition time. However, in the xBD dataset, most of the building objects are low buildings, which weakens the model's ability to detect tall buildings.
- Second, the training samples are imbalanced in terms of damage levels and the disaster categories, as shown in Table 2 and Figure 12. The imbalanced training set may affect the training process of the network. Due to the difficulty in acquiring paired high-resolution pre-disaster and post-disaster remote sensing images, the xBD dataset is the first building damage assessment dataset with high-quality annotations. Therefore, it requires a large amount of work to expand the size of the building damage assessment dataset.

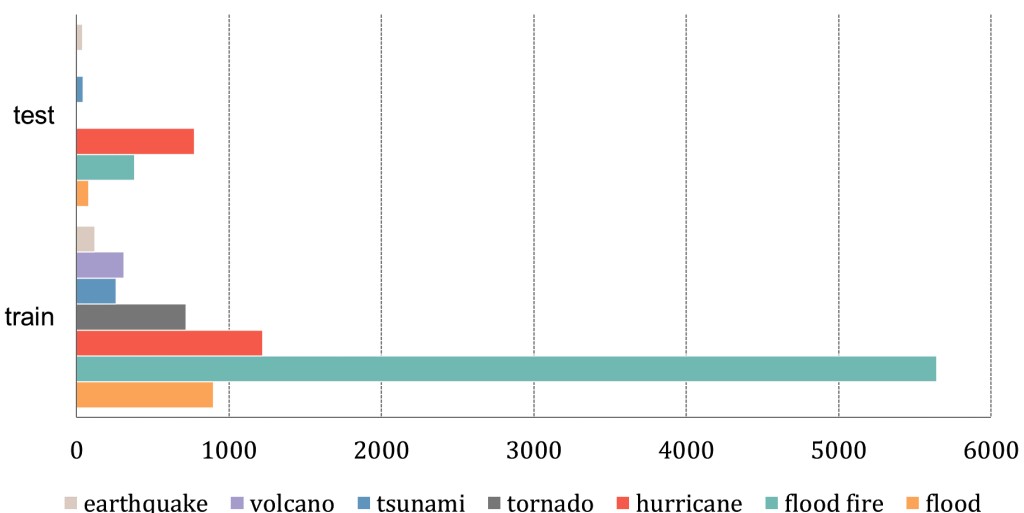

**Figure 12.** The distribution of disaster categories in xBD dataset.

### 4.3. Future Work

We analyzed the advantages and disadvantages of the proposed SDAFormer in detail. In this section, two possible research directions in the future are discussed.

Based on the aforementioned advantages and disadvantages of SDAFormer, further research is needed in the following two aspects.

- First, in future experiments, additional relevant datasets from other remote sensing sensors, including an Unmanned Aerial Vehicle (UAV) system, will be used to train the network. The additional data allow our model to learn more key features and verify the universal applicability of SDAFormer in building damage assessment.
- Second, on a larger scale, the patterns of damaged buildings are more diverse and complex than the existing classification criteria, Joint Damage Scale (JDS) [48], especially for buildings in urban areas. The process of label calibration is subjective, and this certain tendency of labeling can affect the training process. Therefore, more detailed categories of damaged buildings can facilitate the model to analyze more useful information to support HADR. We will further explore the classification method for building damage assessment and the application of Transformer architecture in the field of disasters response.

### 5. Conclusions

In this paper, we propose a two-stage siamese framework based on hierarchical Swin Transformer for building damage assessment tasks named SDAFormer. In Stage 1 of the framework, building segmentation is performed to locate the buildings. Based on the building localization in Stage 1, damage assessment is then performed in Stage 2. Compared with the CNN-based frameworks, SDAFormer can extract long-range semantic information for damage assessment. Moreover, the spatial fusion module is designed to be embedded in the Swin Transformer blocks to facilitate spatial information exchange. SDAFormer is the first to introduce a pure Transformer architecture for a multi-temporal remote sensing interpretation task. Compared with existing CNN-based methods, the proposed framework achieves significant improvements on the large-scale building damage assessment dataset, xBD. Furthermore, four independent disasters are processed for evaluation. The results verify that our framework is robust and has good potential for transferring to other tasks.

**Author Contributions:** Conceptualization, Y.D. and Z.J.; Data curation, Y.D.; Formal analysis, Y.D.; Funding acquisition, Y.Z.; Investigation, Y.D.; Methodology, Y.D.; Project administration, Y.Z.; Resources, Y.D.; Software, Y.D.; Supervision, Y.Z.; Validation, Y.D., Z.J. and Y.Z.; Visualization, Z.J.; Writing—original draft, Y.D.; Writing—review and editing, Y.Z. All authors have read and agreed to the published version of the manuscript.

**Funding:** This research was funded in part by the National Natural Science Foundation of China under Grant 62171016 and 61871413, and in part by the Fundamental Research Funds for the Central Universities under Grant buctrc202001.

**Institutional Review Board Statement:** Not applicable.

**Informed Consent Statement:** Not applicable.

**Data Availability Statement:** lPublicly available datasets were analyzed in this study. This data can be found here: https://www.digitalglobe.com/ecosystem/open-data, accessed on 19 December 2021.

**Conflicts of Interest:** The authors declare no conflict of interest.

## Abbreviations

The following abbreviations are used in this manuscript:

| | |
|---|---|
| HADR | Humanitarian Assistance and Disaster Response |
| SAR | Synthetic Aperture Radar |
| SVM | Support Vector Machine |
| RF | Random Forest |
| CNN | Convolutional Neural Network |
| RNN | Recurrent Neural Network |
| ROI | Region of Interest |
| NLP | Natural Language Processing |
| ViT | Vision Transformer |
| MSA | Multi-head Self-Attention |
| SF | Spatial Fusion |
| MLP | MultiLayer Perceptron |
| LN | Layer Normalization |
| W-MSA | Window-based Multi-head Self-attention |
| SW-MSA | Shifted Window-based Multi-head Self-attention |
| W-Trans | Window-Transformer |
| SW-Trans | Shifted Window-Transformer |
| GSD | Ground Sample Distance |
| TP | True-positive |
| FP | False-positive |
| FN | False-negative |
| UAV | Unmanned Aerial Vehicle |
| JDS | Joint Damage Scale |

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
