# Peer review of "Building Damage Assessment Based on Siamese Hierarchical Transformer Framework"

_mathematics, doi:10.3390/math10111898_

Round 1

Reviewer 1 Report

  1. At the abstract, this is well know given the locality principle, but attention has helped to ameliorate such things.

  2. A review of the English is necessary. For example, “with shallow structures are developed” should be “have been developed”

  3. I noticed that the fusion works as a residual network with a transformer. Is this the inspiration of it?

  4. This is a metric learning type method using a U-Net (Actually you are using contrastive loss), but I do not see any mention of these type of Deep Learners. I believe that it is necessary to talk about them.

  5. A critic Where is your loss function? This is an important part because of the difficulties that such loss function can imply during training (Refer to the practicalities of training GANs). At the loss function is the explanation on how both networks are coupled.

  6. I understand the use of skip connections to generate a type of pyramid of images (Used also in the classic descriptors), but I do not see an attempt to explain theoretically how this fusion structure produce the desired results. Given the level of the journal I would assume that some theorem proofs are necessary.

  7. At page 6-7 there is an attempt to describe Swin Transformer Block, but you are using this as a information bottleneck. There is not a clear explanation of this at the section.

  8. We make the same observation about the fusion module. It lacks theoretical foundations of why this is working.

  9. There is a type of ablation study integrated to the main results when removing the SF module, but it is necessary to have a larger study on this.

Reviewer 2 Report

This manuscript presents a new approach to image analysis in terms of levels of building destruction due to disasters.
The authors use a two-stage analysis method, approaching the problem first by locating buildings, then investigating the level of damage based on the first analysis. Also different functions, modules and structures are used to improve the results of similar studies using classical CNNs.
The results of the applied method are compared with the results of RescueNet and MaskRCNN, providing higher accuracy and lower building recognition errors.
The structure of the manuscript is well written, containing in a logical way the steps of the literature involvement, the research, the application of the method and the results obtained.
The limitations of the method and possible future research directions are also presented.

Strictly considering the proposed method, a more thorough investigation of the results in comparison with other artificial neural network structures is needed. The method also needs to be validated by applying several datasets and perhaps comparing the results with real data measured at the building level. 

Reviewer 3 Report

The authors present in a very well-structured article their new development and its testing, which was created to computerized classification of the damage caused by natural disasters to building systems. The Siamese Hierarchical Transformer Framework, called SDAFormer, evaluates the damage area based on high-resolution satellite imagery taken before and immediately after the damage event, and generally outperforms previous software applications for similar purposes. Although it is structured heavily on systems previously used for similar tasks - Siamese U-Net to perform spatial building layout and classification of damaged buildings, Transformer architecture, MSA function, skip-connections used in Vision Transformer (ViT), and to limit the high computational complexity of ViT the SWin (Shifted Windows) Transformer mechanism for semantic isolation and learning of multilevel features - as well as elements of medical or NLP solutions, completely new ideas were also embodied in its structure. The developed two-stage damage assessment framework embeds a symmetric hierarchical transformer in a Siamese U-Net-like network. The task of the first stage is to determine the location and shape of the building system and to separate it from the environment on the basis of the image reflecting the intact condition. As an ancillary function, the operating network passes its weight values to the damage level classification network operating in the second phase  ​​for initialization. To avoid the flaws of previous CNN-based solutions, it uses a clean Transformer architecture that includes the already known SWin solution. It is more suitable than before for the extraction of temporal and spatial, local and global semantic information carried by the difference between the recordings giving original and damaged condition, and the additional advantage over CNN of the need for linear computation. Own development is the vertical and horizontal self-monitoring mechanism to build pixel-level spatial relationships, thereby increasing the subtlety of visual distinction. Similarly original solution is to use the spatial fusion (SF) module for the global aggregation of spatial features.
The developed, rather complex system is described with the help of excellent, uniform figures.
The xBD data set from the Maxar / DigitalGlobe Open Data Program was used to teach and test the system built using the Pytorch 1.10 Deep Learning modeling framework. This database contains pre- and post-damage recording pairs along with damage classification data. Tests were made with three other systems, as well as their own network without the SF module, with two damage cases, and their SDAFormer system was at the forefront of five of the six damage detection metrics. The comparisons are aided by very spectacular and informative pictures and explanations. The reliability of the system was verified with data from two completely new cases. When evaluating these tests, the effect of the quality of the input images on the results is already mentioned, e.g. different shooting angle. They also talk about the weaknesses of the system, as in the case of high-rise buildings, the deceptive effect of shadows.
Overall, the article demonstrates excellent work. Its shape and content are both excellent. It starts with a very purposeful introduction and literature review. The detailed presentation of the developed system, the demonstration of the training and testing are complete, the figures are excellent. Its language is flawless, maybe a little more concise text would have been enough, when in some cases things were easy to understand. The dissertation also contains a very rich 52-item bibliography and a list of abbreviations.
It is recommended that it be published without reservation.

Round 2

Reviewer 1 Report

Dear authors I am gratefully impressed with your corrections and extensions of your paper. However, If feel that Bottleneck and spatial fusion  process explanation could be improved presenting some Theorem proving why the benefits of using such architectures. If it is possible to include that it would be great.
